# Distribution of calbindin-positive neurons across areas and layers of the marmoset cerebral cortex

**Nafiseh Atapour[1]ᐤ‡, Marcello G. P. Rosa[1]ᐤ‡, Shi Bai[1], Sylwia Bednarek[2], Agata Kulesza[2], Gabriela Saworska[2], Sadaf Teymornejad[1], Katrina H. Worthy[1], Piotr Majka[2]\***

**1** Department of Physiology and Neuroscience Program, Biomedicine Discovery Institute, Monash University, Clayton, Australia, **2** Laboratory of Neuroinformatics, Nencki Institute of Experimental Biology of the Polish Academy of Sciences, Warsaw, Poland

ᐤ These authors contributed equally to this work.
‡ These authors share first authorship on this work.
\* p.majka@nencki.edu.pl

**Data Availability Statement:** The density maps, the segmentations into cortical areas and layers, and exclusion masks due to tissue corruption for each case are available as NIFTI files and can be

## Abstract

The diversity of the mammalian cerebral cortex demands technical approaches to map the spatial distribution of neurons with different biochemical identities. This issue is magnified in the case of the primate cortex, characterized by a large number of areas with distinctive cytoarchitectures. To date, no full map of the distribution of cells expressing a specific protein has been reported for the cortex of any primate. Here we have charted the 3-dimensional distribution of neurons expressing the calcium-binding protein calbindin (CB$^+$ neurons) across the entire marmoset cortex, using a combination of immunohistochemistry, automated cell identification, computerized reconstruction, and cytoarchitecture-aware registration. CB$^+$ neurons formed a heterogeneous population, which together corresponded to 10–20% of the cortical neurons. They occurred in higher proportions in areas corresponding to low hierarchical levels of processing, such as sensory cortices. Although CB$^+$ neurons were concentrated in the supragranular and granular layers, there were clear global trends in their laminar distribution. For example, their relative density in infragranular layers increased with hierarchical level along sensorimotor processing streams, and their density in layer 4 was lower in areas involved in sensorimotor integration, action planning and motor control. These results reveal new quantitative aspects of the cytoarchitectural organization of the primate cortex, and demonstrate an approach to mapping the full distribution of neurochemically distinct cells throughout the brain which is readily applicable to most other mammalian species.

## Author summary

The cerebral cortex is the part of the brain which expanded the most in primate evolution. Part of this change corresponds to differentiation into a larger number of areas, each characterized by a different combination of cells that express different proteins, and the

downloaded from https://www.marmosetbrain.org/whole_brain_cb_maps. Training datasets in the form of annotated counting strips, as presented in Fig 1, are available as TIFF files containing the imaging data and corresponding SVG files containing the annotations. Summarized results that allow one to reproduce the analyses presented in this article are provided in S1–S3 Tables. All data, including training datasets and custom Python scripts for training the Unet CNN can be found at https://www.marmosetbrain.org/whole_brain_cb_maps.

Funding: This study was supported by the National Science Centre grant (2019/35/D/NZ4/03031 to PM), the National Health and Medical Research Council grants (APP1194206 to MGPR) and (APP2019011 to NA). The funders had no role in study design, data collection and analysis, decision to publish, or preparation of the manuscript.

Competing interests: The authors have declared that no competing interests exist.

arrangement of these cells into layers. Finding ways to fully map this diversity has been a challenge. We have developed a workflow based on immunohistochemistry, automated cell identification, and computerized reconstruction which allowed us to map the full distribution of neurons expressing calbindin, a protein that is important for regulating the levels of intracellular calcium. This 3-dimensional map has revealed that cortical areas vary not only with respect to the number of cells expressing calbindin, but also to where they are located. This approach is readily adaptable to mapping the distribution of other proteins, across various species, which will allow future work towards understanding the anatomy, physiology and evolution of the cortex.

## Introduction

One of the essential criteria for characterizing neuronal classes is the cellular expression of molecules such as neurotransmitters, neuromodulatory peptides, and calcium-binding proteins. This type of characterization is often used to distinguish classes of cells that prove, upon additional work, to have distinct functional features and anatomical connections [1–3]. Areas of the cerebral cortex vary in terms of proportions of neurons that express different proteins and their distributions across layers (e.g. [4,5]). However, much research is still needed to fully understand the extent to which areas differ in this respect, and the rules that explain this variety. For example, to date, comprehensive spatial maps of neurons defined by expression of calcium-binding proteins have only been obtained in the mouse brain [6,7]. Our knowledge of this subject in the primate cortex remains less complete, despite recent progress in mapping the different types of cells based on transcriptomics [8,9] and receptor radioautography [10]. A comprehensive knowledge of the distribution of biochemically distinct neurons in non-human primates is important given the marked elaboration of the cortex in primate evolution and their essential role in translational research [11,12].

Here we report on the full distribution of neurons expressing the calcium-binding protein calbindin-D28K (CB$^+$ neurons) in the cortex of the marmoset monkey (*Callithrix jacchus*). Calbindin (CB) can increase the neuronal calcium buffering capacity, which leads to modulatory effects on synaptic plasticity, memory, and other facets of cognition and behavior [13–16]. In addition, it has been suggested that intracellular CB has protective roles which are important in the context of the development of neurodegenerative disorders and recovery from brain injury [17–20]. CB$^+$ neurons in the primate cortex are, in vast majority, inhibitory (GABAergic) interneurons [21–25], but also include, in some areas, a smaller and more variable population of pyramidal cells [26]. The CB marker gene (CALB1) is mainly expressed by the somatostatin (SST) and lysosome-associated membrane protein 5 (LAMP5) subclasses of GABAergic cells, which are among the five main categories of interneurons in the mammalian brain [8,27–29].

Previous studies have described the distribution of CB$^+$ neurons, using either qualitative or quantitative (stereology-based) methods, in specific regions of the primate cortex such as prefrontal [4], temporal [24,26], auditory [30] and visual [25,31] areas. However, we are still lacking a comprehensive spatial analysis of the distribution of CB$^+$ neurons across the entire cortex of any primate. In the present study, the distribution of CB$^+$ neurons across the marmoset cortex was defined using a computational workflow that incorporates artificial intelligence-based identification of neuronal bodies, validated by manual counting by multiple experts, as well as 3-dimensional reconstruction and cytoarchitecture-aware registration [32].

The marmoset is one of the species for which quantitative data on the areal and laminar distributions of cortical neurons are available [33]. This species has also been the subject of large-scale studies focused on structural and functional connectivity [34–38] and of a comprehensive analysis of neuronal types based on single-nucleus RNA sequencing [9]. To allow further studies, the present datasets are also provided for download in a spatially oriented, three-dimensional form (NIFTI files available at https://www.marmosetbrain.org/whole_brain_cb_maps).

## Results

We have applied immunohistochemical techniques to reveal the locations of $CB^+$ neurons (Fig 1A) in the brains of 3 young adult marmosets (one cerebral hemisphere each). From the resulting coronal sections in one of the cases (CJ1741), we selected radially oriented strips from each of the currently recognized cytoarchitectural areas of the cortex in this species (e.g. Fig 1B and 1C). Manual annotation of the positions of $CB^+$ neurons in these strips (Fig 1D and 1E), using the workflow described in [33], resulted in a library of 4,072 counting boxes, which provided the basis for training and evaluation (Fig 2A and 2B) of a U-Net architecture Convolutional Neural Network [39,40] (U-Net CNN, see *Materials and Methods* section). Differentiation between putative subtypes such as those shown in Fig 1A was not attempted; rather, manual annotators were instructed to identify every $CB^+$ cell body.

We found that the U-Net CNN-generated neuronal densities reflect a consensus between the estimates obtained by different human experts (Figs 2C–2E and S3). The U-Net CNN was then used to compute the densities of $CB^+$ neurons in every fifth section across the entire cortex (e.g. Fig 2F and 2G). Adjacent sections stained for Nissl substance or NeuN (Neuronal Nuclear protein; a neuron-specific marker; [41]), myelin and cytochrome oxidase were used to identify cortical areas and layers, as detailed in the *Materials and Methods* section. This allowed the evaluation of the relations between the density of $CB^+$ neurons and categories of areas defined by anatomical-functional, (see hierarchical levels [36,42]) and structural (see type of lamination [4,43]) measures.

### Areal distribution of $CB^+$ neurons

Figs 3 and 4 show the density (neurons·mm$^{-3}$) of $CB^+$ neurons across different cortical areas. For the summary in Fig 3A, cortical areas were arranged in groups defined by location and function [45]. The $CB^+$ neuronal density varies according to area, within an approximately six-fold range ($5 \cdot 10^3$ and $30 \cdot 10^3$ cells·mm$^{-3}$; means for 3 animals; individual data shown in Fig 4). Given that the total neuronal density estimated by NeuN staining is also known to vary substantially between cortical areas [33], a natural question is whether these observations can be explained by a simple model whereby $CB^+$ neurons form a constant fraction of the neuronal population across areas. Our analyses demonstrate that this is not the case. Despite a significant correlation being found between the density of $CB^+$ neurons and total neuronal density (Fig 3B, top left), the data revealed that areas containing higher densities of $CB^+$ neurons also tend to have these forming higher percentages of the total neuronal population (Fig 3B, top right and bottom graphs show data from individual animals). Significant differences between areas in different functional groups persist after correction for total neuronal density (Fig 3C). Thus, different cortical areas show variations in intrinsic cellular circuitry, suggesting different functional requirements mediated by $CB^+$ neurons.

Auditory areas were particularly notable, as a group, for having high absolute and relative proportions of $CB^+$ neurons, with the highest peak relative densities ($CB^+$ neurons as a percentage of the total neuronal population) occurring in the auditory core areas (AuA1, AuR, AuRT). Visual areas also tended to show high absolute $CB^+$ neuron densities (Figs 3 and 4),

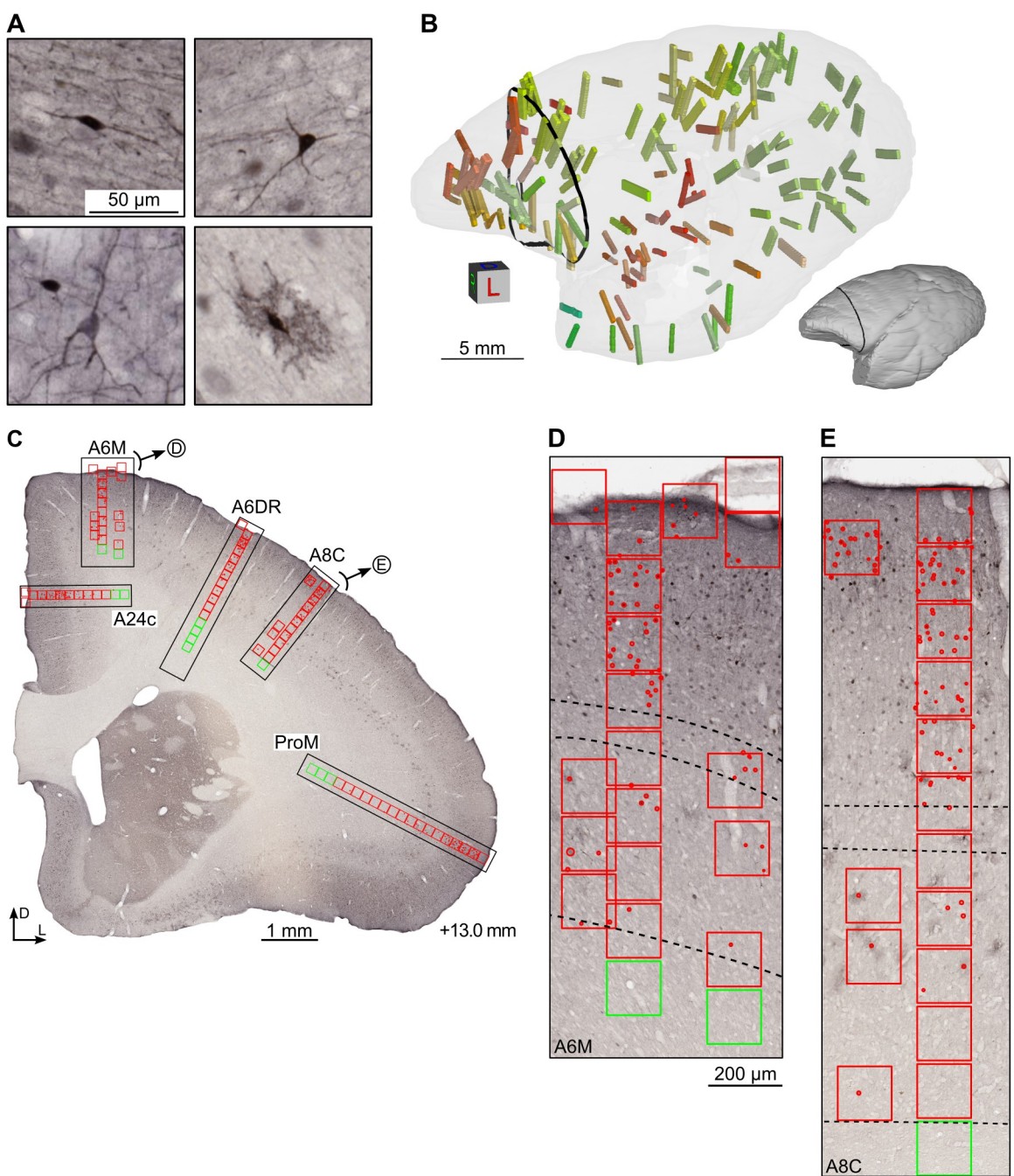

**Fig 1. Manual annotation of calbindin-positive (CB⁺) neurons across the cerebral cortex.** (A) Representative types of CB⁺ neurons (from top left to bottom right): bipolar, multipolar, small pyramidal, and neurogliaform neurons, imaged at ×20 magnification (0.4974 µm per pixel). The neurogliaform neurons were identified according to criteria described in studies of the macaque cortex [22,24], and their neuronal identity has been confirmed by immunofluorescence co-staining with NeuN and GABA in other sections (S1 Fig). (B) Locations of 163 image strips sampled in case CJ1741, visualized against a semi-transparent rostrolateral view of the 3D reconstructed left hemisphere (opaque model presented as a thumbnail for clarity). Different colors of the image strips correspond to the areas they were derived from (see e.g. S1 File), and the black contour indicates the coronal level of the section presented in panel C. (C) Coronal cross-section taken approximately at the interaural +13.0 mm [44] containing five image strips (black rectangles) from areas A24c, A6M, A6DR, A8C, ProM, clockwise. (D, E) Strips from areas 6M and 8C shown at high magnification. Counting boxes are represented with red (gray matter) or green (white matter) squares of 150 µm in size. The thin, dashed band in the middle of the strips indicates the boundaries of layer 4, while the bottom dashed line shows the border between layer 6 and the white matter. The laminar boundaries were determined by comparison with adjacent Nissl-stained sections.

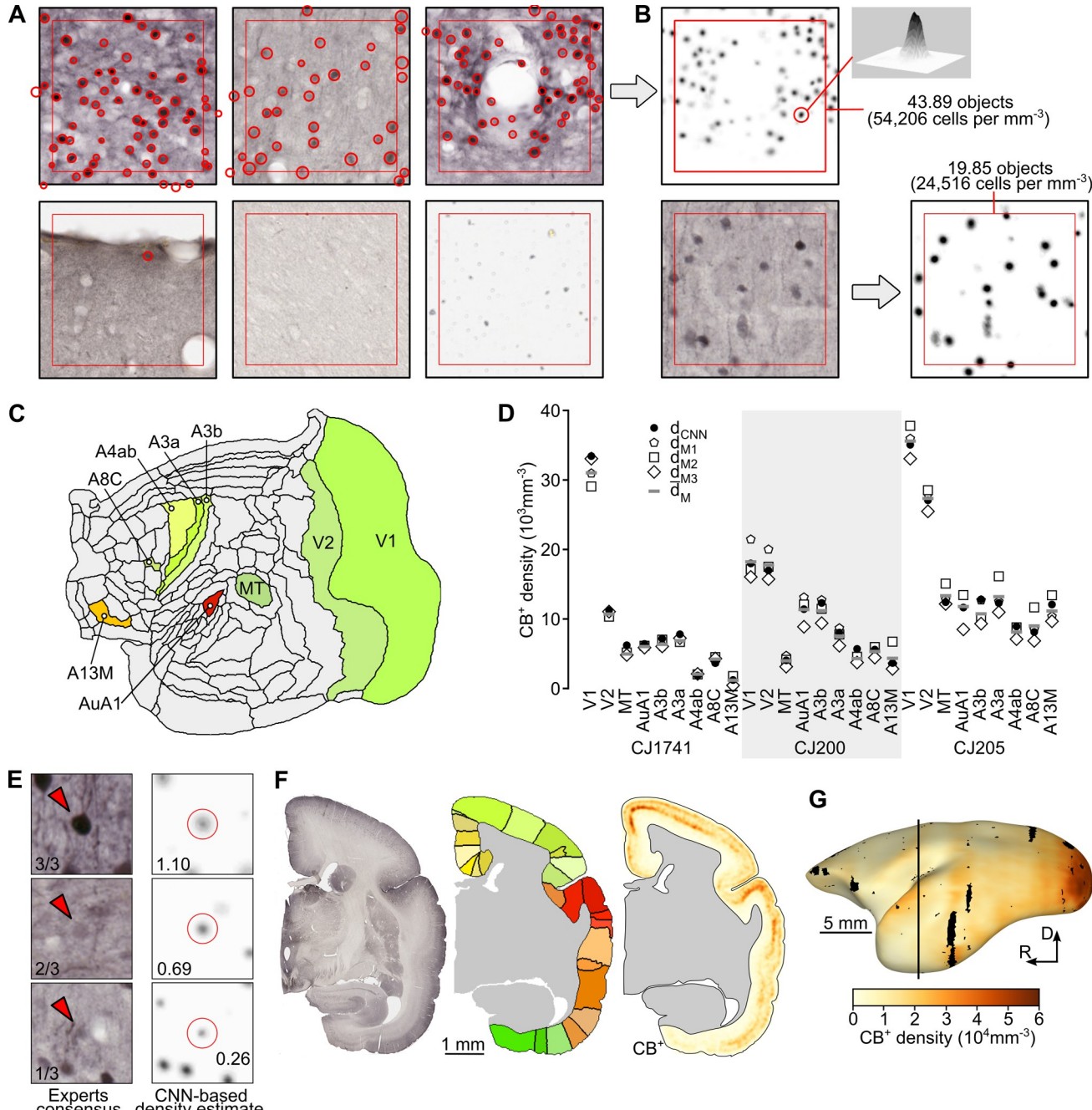

**Fig 2. Training and evaluation of the U-Net Convolutional Neural Network (U-Net CNN).** (A) Examples of counting boxes manually annotated by neuroanatomists (red circles of various sizes). The entire counting box is $150 \times 150$ µm while the area annotated with the red square is a $256 \times 256$ px ($127$ µm $\times 127$ µm) image patch used for training the CNN. A total of 4,072 counting boxes were defined, including boxes representing various densities of neurons *(top left* and *top middle)*, parts of the tissue containing blood vessels or artifacts *(top right* and *bottom left)*, as well as samples of white matter *(bottom center)* or samples that do not depict the brain tissue *(bottom right)*. (B) Density map generated based on the counting box indicated by the gray arrow. A single Gaussian blob corresponds to a single $CB^+$ neuron, and 43.89 neurons are located within the indicated image patch (red square). Note that non-integer cell number estimates are possible, as explained in *Automated detection of $CB^+$ neurons* Methods section. Upon training the U-Net CNN counting boxes not previously presented to the network can be turned into a density maps, and the total number of neurons within an image patch can be computed. (C) Locations of nine areas selected for a comparison between the $CB^+$ densities estimated by the U-Net CNN and multiple human annotators (V1, V2, MT, AuA1, A3b, A3a, A4ab, A8C, and A13M, see S1 File for a list of areas, color coding and abbreviations). (D) Per-area (i.e. average values for all boxes sampled from a given area) densities for the three analyzed hemispheres. Different symbols show the results for individual human experts ($d_{M1}$ to $d_{M3}$), an average of the three expert observers ($\bar{d}_M$), and the densities obtained with the U-Net CNN ($d_{CNN}$). The mean of the differences between $d_{CNN}$

and the $d_M$ densities is statistically indifferent from zero (see S3F and S3G Fig for statistical details), indicating that the automatic and the average manual counts are indistinguishable. (E) The results obtained by the U-Net CNN reflect the consensus between the expert annotators. (*rows, top to bottom*) Examples of individual CB[+] neurons marked by all, two, and only one expert, respectively, and a proportional density estimate by the U-Net CNN. The proposed method helps alleviate the interindividual variability of manual cell counting. Box size: 50 μm. (F) We applied the procedures for estimating the density of CB[+] neurons to all CB-stained sections in all three hemispheres studied. Here, results for an example section (CJ1741-r16c) are presented. From left to right: section image, segmentation of the cortex into individual areas based on manual parcellation and coregistration to the reference template [35] (see *Materials and Methods* for details), and density map of CB[+] neurons (see G for scale). Note that the quantification of the results is performed only in the cortical areas, while the subcortical regions are not considered. (G) Example three-dimensional reconstruction of a CB[+] density map constituting the basis of the flat maps illustrated in other figures. The black line indicates the location of the section presented in panel F, and the black patches correspond to the parts of cortex excluded from the analyses due to staining artifacts or corrupted tissue. The datasets are available for download from https://www.marmosetbrain.org/whole_brain_cb_maps.

but these are somewhat offset by high overall neuronal densities [33], resulting in less notable relative densities (Fig 3C). The primary visual cortex (V1) was an outlier both in terms of overall CB[+] neuronal density, and in having these neurons forming a very high proportion of the total neuronal population (Fig 3B and 3D; see Fig 5 for individual data), with the second visual area (V2) providing a transition between the extreme values in V1 and those in other visual areas. Although the somatosensory cortex (SSC) included areas with a wide range of CB[+] neuronal densities, the primary areas (areas 3a and 3b) represented well-defined local maxima, which contrasted with the adjacent motor/ premotor complex and posterior parietal areas (Figs 3D and 5).

Although isocortical association areas did not show extreme variations in CB[+] neuronal density, a finer-grained analysis revealed a few notable trends. For example, in the prefrontal cortex, ventrolateral prefrontal areas (VLP) such as 45, 47M, and 47O tended to show higher percentages of CB[+] neurons in comparison with other (frontopolar, dorsolateral prefrontal, orbitofrontal, and medial prefrontal) subdivisions (e.g. Fig 3C). In the temporal lobe, caudal lateral and inferior temporal areas such as TEO, TE2, and TE3 showed higher absolute CB[+] neuron densities in comparison with ventral (VTC) areas such as 36 and TH (Fig 3A), but relative densities showed less variation (Fig 3C). The posterior parietal areas (PPC) were distinct, as a group, for low relative densities of CB[+] neurons (Fig 3C).

## Correlation between the distribution of CB[+] neurons and hierarchical levels

One of the basic organizational principles of the primate cortex is the existence of an anatomical hierarchy of areal connectivity, which can be quantified based on the laminar origins and terminations of cortico-cortical connections [46,47]. This hierarchy, which is theorized to reflect the principal direction of information flow from sensory input to the generation of behavioral responses and executive function [42,48,49] has been determined for many areas of the marmoset cortex [36]. This raises the question of whether there is a systematic relation between the density of CB[+] neurons and the hierarchical rank. We found that hierarchically "low" areas, such as the primary sensory areas, tend to show high densities of CB[+] neurons, compared to "high" areas (Fig 3E). A similar but weaker relation is also detected when the percentages of CB[+] neurons in different areas was considered (Fig 3F).

Dombrowski et al. [4] demonstrated that the distribution of cell types defined by expression of calcium-binding proteins varied according to the degree of differentiation between cortical layers (a characteristic that is hypothesized to reflect the evolutionary history of the mammalian cortex [50]) in areas of the macaque prefrontal cortex. When analyzed according to the six main cytoarchitectural categories of lamination proposed for the primate cortex (adapted for the marmoset cortex classification detailed in [43]), we found that CB[+] neurons tended to

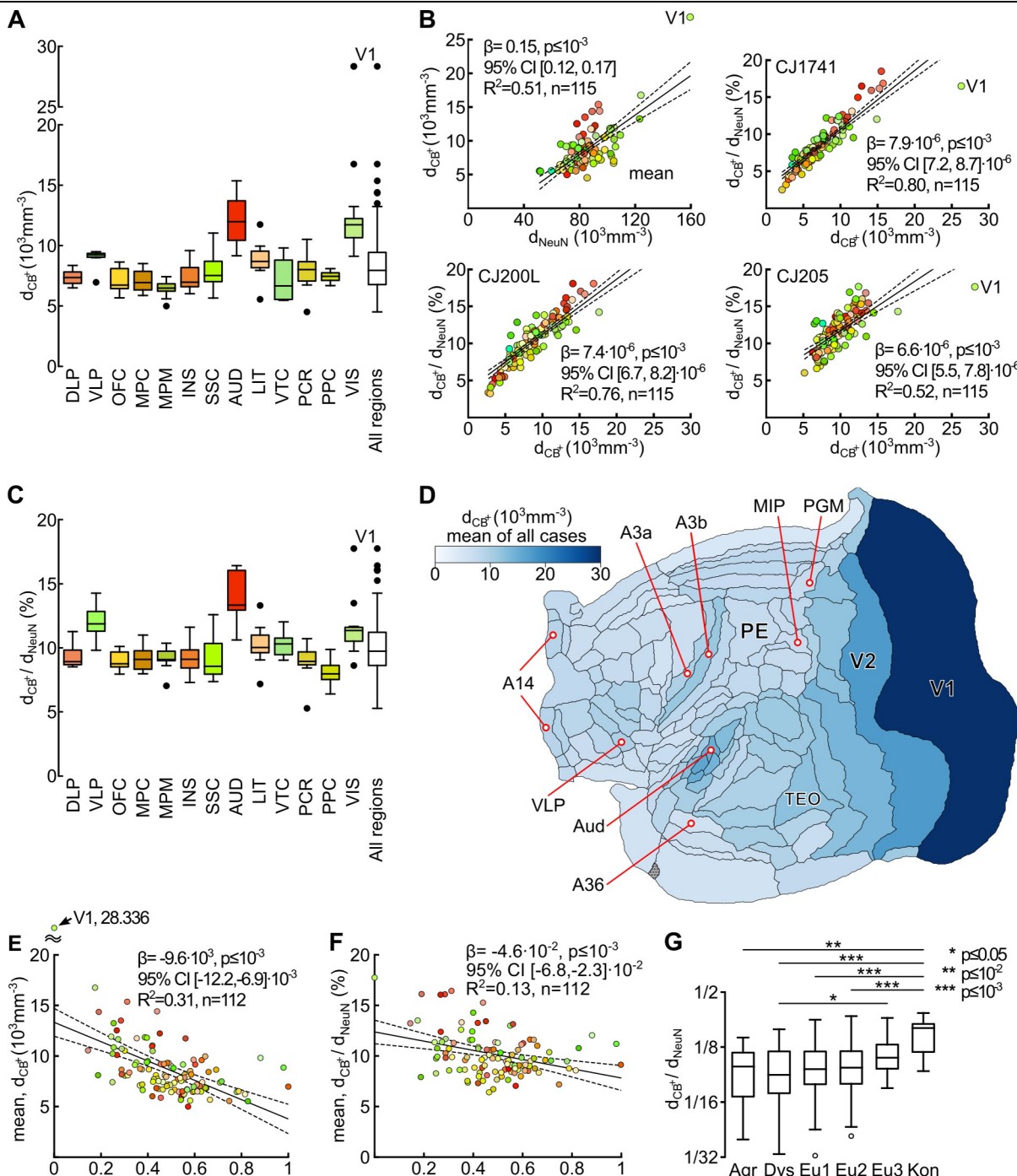

**Fig 3. Variation in the density of calbindin-positive (CB⁺) neurons across the marmoset cortical areas.** (A) Density (neurons·mm⁻³) of CB⁺ neurons across areas, grouped according to the classification proposed by [45]. For each box plot, the center line indicates the median, the box limits the upper and lower quartiles, the whiskers 1.5× the interquartile range, and the annotated points outliers). Abbreviations: DLP: dorsolateral prefrontal cortex; VLP: ventrolateral prefrontal cortex; OFC: orbitofrontal cortex; MPC: medial prefrontal cortex; MPM: motor and premotor cortex; INS: insular cortex; SSC: somatosensory cortex; AUD: auditory cortex; LIT: lateral and inferior temporal cortex; VTC: ventral temporal cortex (encompassing parahippocampal, perirhinal and entorhinal areas); PCR: posterior cingulate and retrosplenial cortex; PPC: posterior parietal cortex; VIS: visual cortex. (B) Top left: the relation between the absolute density of CB⁺ neurons and total neuronal density (NeuN staining) in different areas. Top right and bottom: relations between relative and absolute densities of CB⁺ neurons in 3 animals. (C) Percentages of CB⁺ neurons across groups of areas, classified as in panel A. (D) Flat map representation of the CB⁺ neuronal density (mean of all 3 animals) in different areas. Some of the cortical areas are identified for orientation. (E, F) The relation between the absolute and relative densities of CB⁺ neurons and hierarchical level derived from laminar patterns of connections between cortical areas [36]. (G) Differences in relative density of CB⁺ neurons between areas according to the degree of lamination (adapted from [43]). Abbreviations: Agr: agranular areas; Dys: dysgranular areas; Eu1, Eu2, Eu3: eulaminate areas with increasing levels of laminar differentiation; Kon: koniocortical areas. Colors in B, E, and F correspond to those in A and C; see S1 File for a full list of areas, color coding and abbreviations.

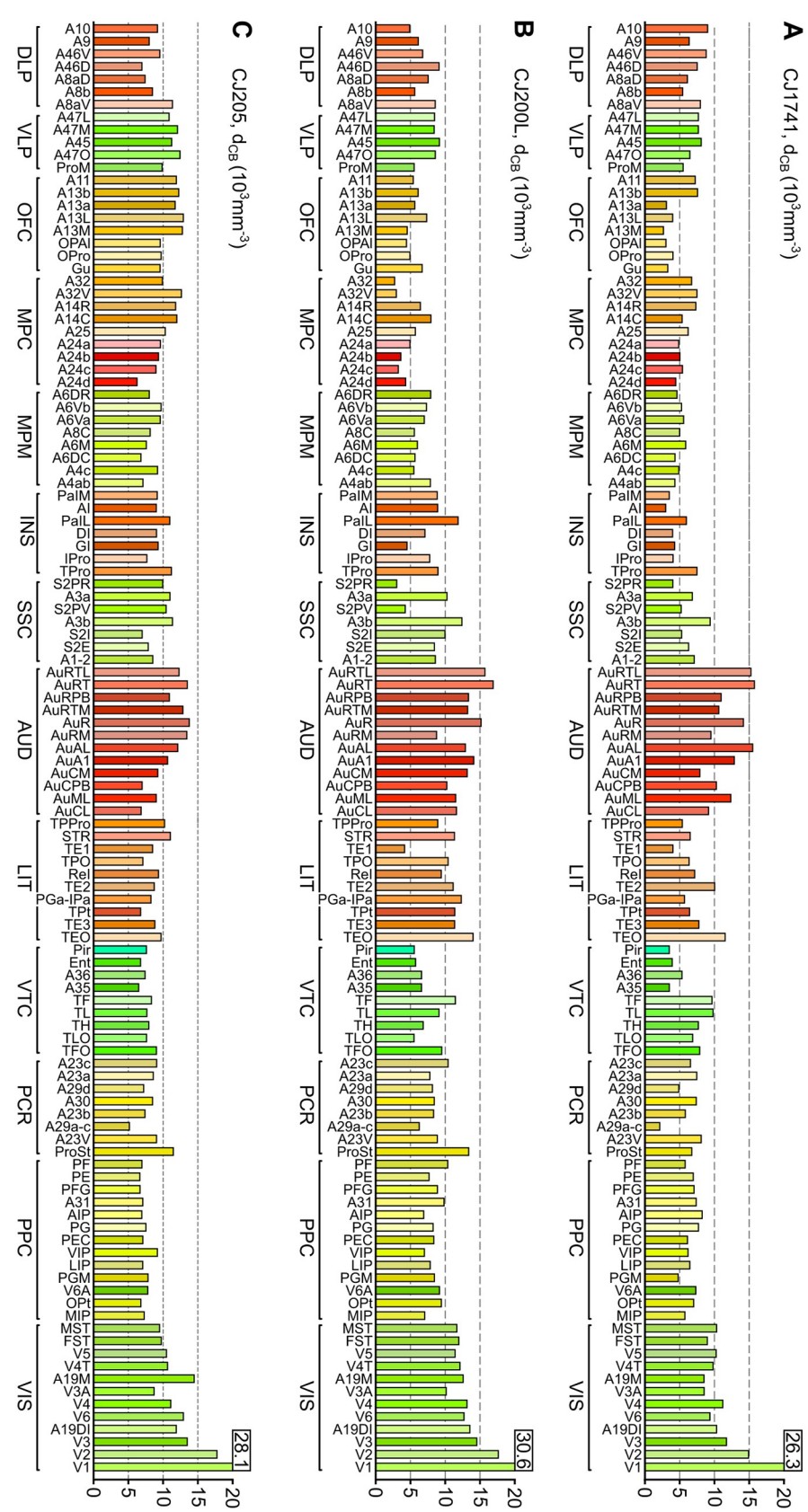

**Fig 4. Densities of CB+ neurons in cortical areas of three marmosets.** For this figure, areas were grouped according to the classification proposed by [45] (modified from [44]). Abbreviations (groups of areas): DLP: dorsolateral prefrontal cortex; VLP: ventrolateral prefrontal cortex; OFC: orbitofrontal cortex; MPC: medial prefrontal cortex; MPM: motor and premotor cortex; INS: insular cortex; SSC: somatosensory cortex; AUD: auditory cortex; LIT: lateral and inferior temporal cortex; VTC: ventral temporal cortex (encompassing parahippocampal, perirhinal and entorhinal areas); PCR: posterior cingulate and retrosplenial cortex; PPC: posterior parietal cortex; VIS: visual cortex. See S1 File for a full list of areas, color coding and abbreviations.

form a higher proportion of the neuronal population in areas with the highest degree of lamination (eulaminate 3 and koniocortex), but few differences were apparent between other types of the cortex (Fig 3G).

## Laminar distribution of CB+ neurons

Next, we addressed the extent to which the laminar distribution of CB+ neurons varied in different areas. To perform this analysis, the upper and lower boundaries of layer 4 were delineated in adjacent Nissl-stained sections, generating a mask that was transferred to the 3-dimensional reconstruction based on CB-stained sections. In a few isocortical areas where a clearly delineated layer 4 was not obvious (e.g. the representations of the limb and axial musculatures in the primary motor cortex, A4ab), we analyzed separately a thin (75 μm wide) strip of cortex centered on the interface between layers 3 and 5; areas without a defined layer 4 homolog (e.g. the entorhinal and piriform areas, see *Materials and Methods*) were not considered in this analysis. Other landmarks including the interfaces between layers 1 and 2 and between layer 6 and white matter were also incorporated, therefore allowing us to analyze the distribution of neurons according to three compartments in each area: layers 2 and 3 (supragranular), layer 4 (granular) and layers 5 and 6 (infragranular).

Although in general CB+ neurons are heavily concentrated in the supragranular and granular layers of the cortex (S2 Fig), significant quantitative differences were observed between areas, as detailed below. To investigate these areal differences, we calculated two ratio-based measures: the ratio of CB+ neuronal density in supragranular to infragranular layers ($d_{2,3}/d_{5,6}$ ratio; Fig 6A and 6C), and the ratio of CB+ neuronal density in supragranular to granular layers ($d_{2,3}/d_4$ ratio; Fig 6B and 6D).

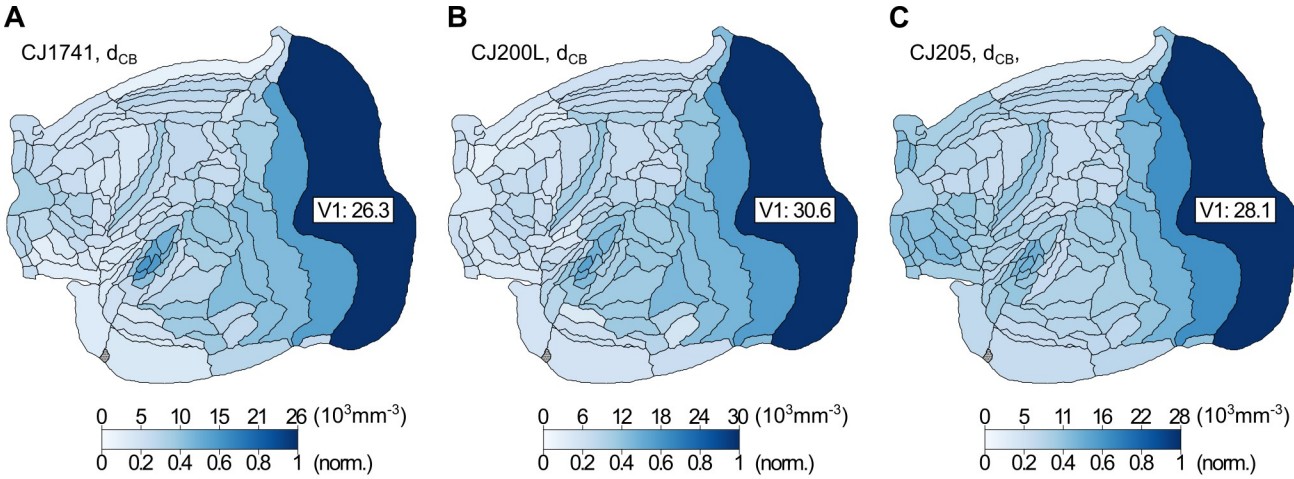

**Fig 5. Maps of the distribution of CB+ neurons in three animals, shown in unfolded representations of a left hemisphere.** For each map, the upper scale indicates densities in neurons·mm$^{-3}$, and the lower scale shows the same data normalized to the peak density of a given individual.

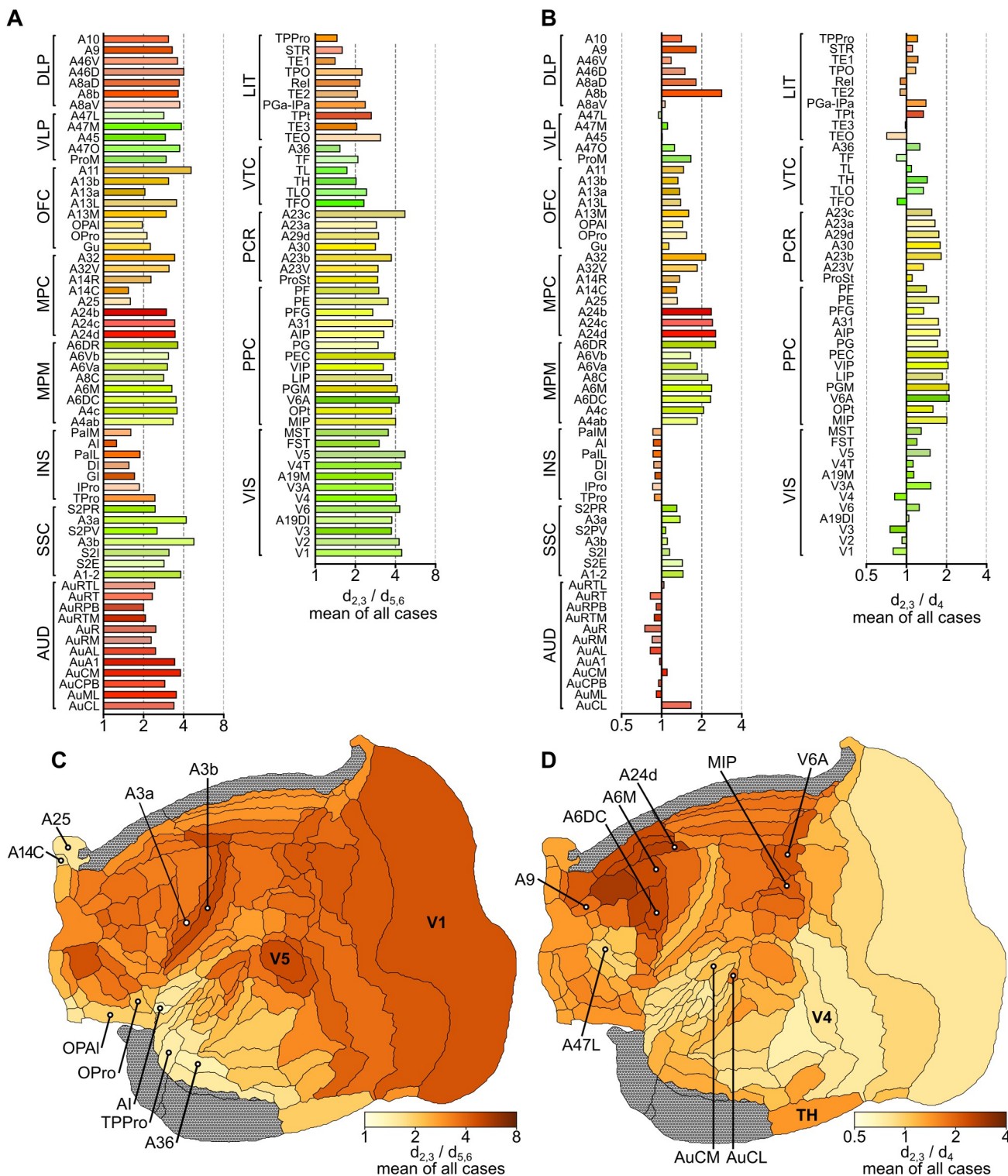

**Fig 6. Laminar trends in the distribution of calbindin-positive (CB⁺) neurons in areas of the marmoset cortex.** (A) Ratios of the density of $CB^+$ neurons in supragranular versus infragranular layers ($d_{2,3} / d_{5,6}$). (B) Ratios of the density of $CB^+$ neurons in supragranular layers versus layer 4 ($d_{2,3} / d_4$). (C, D) Flat map representations of the distributions of $d_{2,3} / d_{5,6}$ and $d_{2,3} / d_4$, summarizing the data shown in (A) and (B). Areas without a clear homolog of layer 4 were excluded from analysis, indicated in gray.

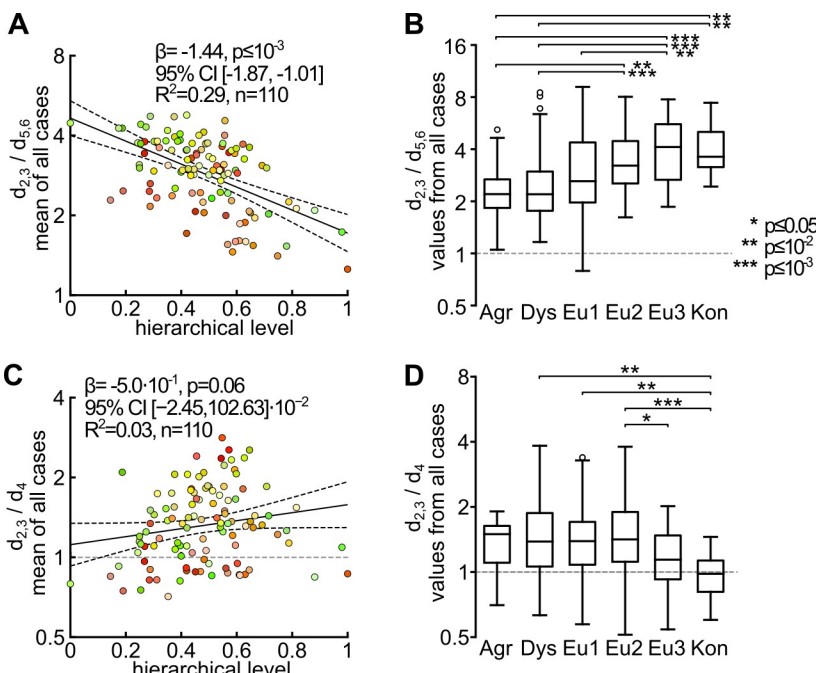

**Fig 7. Relations between the laminar distribution of CB$^+$ neurons and large-scale trends in cortical organization.**
(A) The $d_{2,3}$/ $d_{5,6}$ ratios were strongly correlated with the area's hierarchical level, derived from the laminar pattern of connections. This demonstrates that the bias towards higher density of CB$^+$ neurons in supragranular layers was most marked in low-level (e.g. purely sensory) areas, and more subtle in association and premotor areas. (B) In parallel, the $d_{2,3}$/ $d_{5,6}$ ratios varied systematically with respect to the classification of an area according to the degree of laminar differentiation. Analyses of the laminar distribution of CB+ neurons. (C) In contrast with (A), there was no significant correlation between $d_{2,3}$/ $d_{L4}$ and hierarchical level. (D) Analysis of $d_{2,3}$/ $d_4$ relative to the degree of cortical lamination indicated significantly lower $d_{2,3}$/ $d_4$ ratios (i.e. higher relative concentrations in layer 4) in koniocortex and eulaminate 3 areas, compared to others.

As shown in Fig 6A, the ratio of CB$^+$ neuronal density in supragranular to infragranular layers ($d_{2,3}$/ $d_{5,6}$ ratio) was >1 in every area. However, the data also revealed significant differences between areas, with this ratio varying between 1.28 (agranular insula, AI) and 4.88 (area 3b). Lower $d_{2,3}$/ $d_{5,6}$ ratios (<2) were typically associated with areas of the limbic cortex, including most subdivisions of the insula (e.g. the agranular insula, AI), caudal orbitofrontal cortex (e.g. OPAl, OPro), ventromedial frontal cortex (areas 14C and 25), the temporal pole (TPPro) and ventral temporal areas (e.g. area 36). In contrast, higher ratios (strong supragranular bias) were found in most visual areas and in primary somatosensory areas (A3a and A3b), among others. Most of the areas of association isocortex and motor/ premotor areas (MPM) had intermediate $d_{2,3}$/ $d_{5,6}$ ratios, with CB$^+$ neurons being 3–4 times as densely distributed in the supragranular layers, compared to infragranular layers. In addition, there was a strong negative correlation between CB$^+$ neuron $d_{2,3}$/ $d_{5,6}$ ratio and hierarchical level (Fig 7A), as well as a clear trend for an increase in this ratio with an increasing degree of lamination (Fig 7B).

There were, in addition, systematic differences between areas in terms of the ratio of CB$^+$ neurons densities in layers 2–3 and layer 4 ($d_{2,3}$/ $d_4$). Here, the data revealed that in visual, auditory, and somatosensory areas the density of CB$^+$ neurons in layer 4 tended to be similar to that found in the supragranular layers (Fig 6B) resulting in relatively low $d_{2,3}$/ $d_4$ ratios. In contrast, CB$^+$ neurons tended to be sparsely distributed around layer 4 in premotor areas (e.g. A6M, A6DC), anterior cingulate (medial motor) areas (e.g. A24d), and posterior parietal areas

that have known functions in visuomotor and somatomotor planning (e.g. MIP, V6A), resulting in much higher $d_{2,3}/d_4$ ratios. This functional association was also suggested when visual and auditory areas belonging to different functional streams were considered. For example, visual areas linked primarily to the ventral stream (e.g. V4, TEO, TE3) all had $d_{2,3}/d_4$ ratios $<1$, thus resembling V1 and V2 in having relatively high densities of $CB^+$ neurons in layer 4 (Fig 6B). In contrast, dorsal stream areas (e.g. V3A, V5, V6, MST, and FST), which provide critical input for the frontoparietal network (and thus for visual control of action [51,52]), were all characterized by $d_{2,3}/d_4$ ratios $>1$. Following a similar trend, whereas most of the auditory cortex areas had $d_{2,3}/d_4$ ratios below 1, the exceptions were caudal auditory belt areas (AuCL, AuCM), which are part of the auditory dorsal ("where") stream [53]. An analogous functional distinction in $CB^+$ neuron distribution extended even to the prefrontal cortex, where ventrolateral areas (e.g. subdivisions of area 47), theorized to represent extensions of the visual and auditory ventral streams [54,55], all showed low $d_{L23}/d_{L4}$ ratios in comparison with dorsolateral areas associated with the dorsal streams (e.g. areas 8b, 8aD, 9 and 46; see [56]).

The correlation of $d_{2,3}/d_4$ with an area's level in the anatomical hierarchy was not significant (Fig 7C). Likewise, an analysis relative to the degree of cortical lamination indicated only a few differences, with the exception of lower $d_{2,3}/d_4$ ratios (i.e. higher relative concentrations in layer 4) in koniocortex (Fig 7D). In summary, the relative paucity of $CB^+$ neurons in layer 4 appears to be more correlated with an area's function (in particular its involvement in sensorimotor integration or planning of action) rather than its relation to structural attributes of the cortex.

## Discussion

### Significance for the field of Neuroinformatics

The incorporation of neuroinformatics to the discipline of neuroanatomy has been most evident in the rapid progress in our knowledge of the human brain, including the combination of results from non-invasive techniques which have allowed new insights at millimeter to centimeter scales [57]. More recently, comparable change has been taking place in our knowledge at the level of cells and their connectivity, primarily using animal models. Starting with studies in rodents [6,7,58], the traditional approach of investigating areas and nuclei one or a few at a time is being complemented by projects involving more comprehensive datasets. Such approaches have recently made their way to the brains of non-human primates, revealing new information based on single-neuron resolution connectivity [35,59] and transcriptomics [8,9,60], sometimes combined with non-invasive imaging [34,38]. However, one missing link in these efforts has been their relation to neurons defined by biochemical identity, for which a wealth of cytological and physiological knowledge is available. The present paper demonstrates a strategy to overcome this gap. Using a combination of traditional immunohistochemistry, expert annotation, and neuroinformatics solutions (the latter aimed at high-throughput image analysis, brain reconstruction and cytoarchitecture-aware registration), we demonstrate the feasibility of obtaining maps of the distribution of neurons showing preferential expression of a calcium-binding protein across the entire marmoset cortex.

### Estimating cellular densities: regression counting vs instance segmentation methods

The *density* or *regression counting* paradigm employed in this study allows for estimating the total number of labeled cells within a defined region (e.g. a counting box) instead of identifying individual instances of $CB^+$ neurons. This approach significantly reduces the annotation time,

which is beneficial in situations when objects are densely packed, there are large expanses of tissue is to be analyzed, or the imaging data has a resolution insufficient to visualize the details of neuronal morphologies. From this perspective, the proposed approach is affordable in terms of ease of implementation, modest annotation workload, and fast execution in production. Therefore, it can be more easily applied to the brain in less commonly used species without the requirement of genetic modifications [6], or expensive large-scale spatial transcriptomics [8], hence facilitating studies focused on comparative anatomy. Another advantage is that, by relying on the strong expression of a protein, it can be related to traditionally recognized cell types, about which there is a wealth of physiological information that can be leveraged for biophysical modeling [61–63].

Expert annotators exhibit a natural bias [64,65] and their annotations vary between each other or over time (i.e. between plotting sessions). Our approach, however, represents a consensus among human annotators in the dataset used in this study (as shown in Fig 2D and 2E), thereby helping to mitigate these biases. Evaluation of the model's performance showed that the discrepancy between the manual and U-Net CNN counts reduces with the amount of tissue under consideration (S3F and S3G Fig, see *Materials and Methods*, *Comparison against multiple human raters)*.

Alternative approaches could include object detectors based on classical machine learning methods, such as random forests [66,67] or deep learning instance segmentation architectures. In particular, transformer–based algorithms [68] such as MEDIAR [69], are increasingly used in similar contexts over the recent years. These solutions use self-attention mechanisms to generalize better on diverse datasets thus require relatively limited transfer learning. Transformer–based approaches seem to be an advancement over algorithms relying predominantly on the convolutional neural networks (CNNs) such as Cellpose [70] or DeepLab [71]. The latter exhibit relatively limited generalization capability and typically require extensive and well-curated outlines of individual cells, which could potentially offset the cost reduction promised by the machine learning approaches in the first place (see [72] for a comprehensive review).

To encourage the development of improved methods for identifying and exploring $CB^+$ density patterns, we have released the dataset used to train our model at https://www.marmosetbrain.org/whole_brain_cb_maps. It includes individual counting strips and annotations into boxes and marked cells (Fig 1B–1E). Finally, we provide the results not only in tabular, but also in three-dimensional form. This enables much wider interoperability, for instance warping the $CB^+$ neuronal density maps into the space of MRI marmoset brain templates [34], and analyzing the density patterns in the context of structural and functional connectivity.

## Mapping cell distributions across the entire cortex

The present method has the capacity to obtain maps of the distribution of cells showing the expression of a protein for the whole cortex of single individuals, hence robust detection of quantitative differences between areas. Among other reasons, this is significant given that there can be method-related differences in estimates of the prevalence of the same biochemically-defined cell type obtained by different laboratories, in different individuals [70]. Moreover, although the estimates of absolute cell densities can vary between individuals, the relative density demonstrates a highly reproducible pattern (Fig 5). It remains unclear whether individual variation in the absolute density of $CB^+$ neurons reflect methodological factors, or genuine (biologically relevant) differences, such as those related to sex [73], age [74]or postnatal experience. This is an important question that will require systematic analysis of a much larger number of individuals, since simple measures of dispersion seem insufficient to address this matter (S6C-S6E Fig).

The only previous study which has attempted an extensive mapping of CB$^+$ neurons in the cortex was focused on the mouse brain [7]. The distribution reported in this species is significantly different from the present results obtained in a non-human primate. For example, a significant peak in CB$^+$ neuronal density was observed in the infralimbic area, which was not apparent in the corresponding region of the marmoset brain (subdivisions of medial prefrontal cortex). In addition, particularly low densities were observed in retrosplenial and posterior cingulate areas, which were unremarkable in this respect in the marmoset brain. Furthermore, the densities in auditory, visual and somatosensory areas were similar in the mouse, in contrast with differences observed in the marmoset. However, in both species relatively low densities of CB$^+$ neurons were observed in motor and premotor areas. Finally, the absolute CB$^+$ neuron densities were much lower in the mouse cortex (~300–3500 cells·mm$^{-3}$) compared to marmoset (~5000–25000 cells·mm$^{-3}$) or estimates obtained in the macaque frontal lobe [4]. These results underline the importance of comparative approaches studies at understanding the diversity of the cortex across species, which the present methodology enables.

Although there have been previous reports of diversity in CB$^+$ neuron distribution across areas of the primate cortex [4,22,25,75], the present results establish that these differences are not only of a larger magnitude, but also systematic in ways not previously appreciated. For example, auditory areas are notable in their high densities of CB$^+$ neurons, and there were cortex-wide correlations between the prevalence of these cells and both hierarchical levels, defined by laminar pattern of connections, and type of lamination. These observations, which do not simply reflect differences in overall neuronal density, are compatible with the notion that sensory cortex (in particular, primary sensory areas) differs from association and motor planning areas in terms of intrinsic circuitry, including the required levels of GABAergic inhibition (one of the main functions associated with several types of CB$^+$ neurons).

## Differences in laminar distribution

This heterogeneity in neuronal circuitry is also underlined by systematic differences in laminar distribution. Although CB$^+$ neurons are overall more densely distributed in the supragranular and granular layers, this trend was more pronounced in sensory areas, and least pronounced in limbic areas, in correlation with hierarchical levels. Given the well-documented differences in the distribution of projection neurons forming feedforward and feedback connections (with the latter being formed primarily by infragranular pyramidal cells [42,48]), this result suggests that the distribution of CB$^+$ interneurons reflects to some extent functional requirements linked to the physiological shaping of the interareal projection output. In addition, across the entire cortex areas involved in sensorimotor integration and action planning showed reduced densities of CB$^+$ neurons in layer 4, compared to areas more directly related to sensory analysis. This correlation is independent of the hierarchical levels at which such function is performed, or the presence of a clearly defined layer 4 (e.g. it is also evident in posterior parietal areas). These observations may reflect the temporal level of integration needed for different functions [76], with the shorter time scale required for visual and auditory analysis requiring more precise spatiotemporal modulation of the levels of inhibition at the level of the sensory input layers. Together, the present results significantly extend our knowledge of the degree of heterogeneity across areas of the primate cortex, which will need to be considered by future models of cortical function.

In the present study the assignment of CB$^+$ neurons to layers was based on superimposition of CB-stained sections upon adjacent Nissl-stained sections. The use of Nissl for layer delineation reflects the fact that this stain has been used in most cytoarchitectural studies to date, therefore providing a standard reference translatable to most species (despite the fact that there may be discrepancies between layer delineations provided by different stains, and variations according to

stage of development [77]). Although the superimposition was always found to be straightforward, it was a time-consuming step. Contingent on the availability of large-volume fluorescence scanning, further refinement of the method could include the use of DAPI staining to reveal a Nissl-like pattern of layers [78] in the same sections containing antibody-labeled neurons.

## Limitations of the study

The primary limitation of the present approach is that it cannot differentiate between neurons that share preferential expression of a protein but may be physiologically distinct. The focus of the present study was to establish density maps of $CB^+$ neurons rather than analyze their subtypes. However, this limitation can be at least partially addressed in future work by using deep learning architectures capable of instance segmentation [79,80] to distinguish morphological categories [81]. Additionally, adopting higher imaging magnifications [82] combined with double-labeling for other molecular markers and followed by generating expert-curated training sets, could provide further insights on the distribution of more specifically defined cell categories.

Despite carefully executed experimental protocols, the staining intensity variation is an unavoidable part of immunohistochemical staining, particularly when a large number of sections is considered, such as in this study (between 146 and 156 sections, depending on the case). This variation manifests in vertical bands in the 3D reconstruction of the brain hemisphere (e.g. S5B Fig) and could potentially affect the density estimates. The primary means of mitigating this effect was to make the model resilient to the staining intensity variation by establishing a diverse training dataset (Fig 2A) and applying extensive image augmentation during the training. While this resulted in robustness against variations in staining intensity that is comparable to that achieved by human experts (S4 Fig), some artifacts in density estimation due to the slicing plane were still evident (Fig 2G). Given that the quantitative analyses we present are based on averages across several coronal sections, and the random nature of such artifacts, this factor is unlikely to have affected our main conclusions.

Our analysis of laminar variations was based on delineation of three compartments, namely the supragranular, granular and infragranular layers. This level of compartmentalization was chosen as it allowed a high level of consistency in the analysis despite cytoarchitectural variations across areas (for example, the existence of finer subdivisions of layer 3 in some areas, and indistinct boundaries between layers 5 and 6 in other areas). In addition, it allowed direct comparison with connectivity-based estimates of other measures of cortical network structure, which are usually expressed in terms of a ratio of labeled neurons in supragranular vs. infragranular layers [46–48]. Finer differentiation between layers by future studies may reveal other trends and features in the distribution of $CB^+$ neurons. In addition, the fact that the analysis required identification of layer 4 as a necessary step meant that areas where the precise homolog of this layer is unclear (gray regions in Fig 6) had to be excluded.

## Conclusions

Calbindin has long been recognized as one of the main calcium-binding proteins in the brain, but there has been no comprehensive mapping of cells expressing this protein across the entire primate cortex. Enabled by computational techniques, the present results highlight a previously unsuspected degree of heterogeneity across cortical areas and layers, and significant differences relative to data obtained in rodent brains. The methods developed for the purpose of this study are generally applicable to the distribution of other cells that can be revealed by immunocytochemistry, hence offering a flexible solution for whole-cortex mapping.

## Materials and Methods

### Ethics statement

This project was reviewed, approved, and monitored by the Monash University Animal Ethics Committee (Project ID: 26071 "Enabling an accurate model of brain wide connections").

### Experimental model and subject details

After experiments involving retrograde tracer injections unrelated to the present study, three marmosets (*Callithrix jacchus*, aged between 41–43 months, including two females [CJ1741 and CJ200] and one male [CJ205]) were overdosed using sodium pentobarbitone (100 mg/kg, i.v.), and perfused transcardially using heparinized saline, followed by 4% paraformaldehyde in 0.1 M phosphate buffer (PB). The collected brains were post-fixed overnight and cryoprotected in a buffered paraformaldehyde solution containing increasing concentrations of sucrose (10%, 20%, and 30%) over a period of a week. The hemisphere without injections (right hemisphere for CJ1741 and CJ205; left for CJ200) was sectioned at 40 μm, yielding five sequential series of coronal sections used for different stainings. Myelin and Nissl-stained series of sections were used to delineate area boundaries (see [33,35] for details), with the Nissl-stained sections being in addition used to define the cortical layers (see below). Another series was used for CB immunostaining, as described previously [83]. Briefly, the sections were incubated in blocking solution (10% normal horse serum and 0.3% Triton-X100 in 0.1 M PB) for one hour at room temperature before undergoing primary antibody (calbindin-D28K, 1:8000, Swant Swiss, Code No. 300, RRID: AB_10000347) incubation at 4˚C. A biotinylated horse anti-mouse IgG secondary antibody (1:200, PK-6102, Vectastain elite ABC HRP kit, Vector Laboratories, RRID: AB_2336821) incubation was then conducted for 30 min—followed by treatment with Avidin-Biotin Complex (ABC) reagent (PK-6102) and DAB substrate working solution (DAB kit SK-4100, RRID: AB_2336382) as a chromogen. Stained sections were scanned using an Aperio Scanscope AT Turbo system (Leica Biosystems) at ×20 magnification, providing a resolution of 0.4974 μm per pixel.

### Method details

**Manual annotation of CB⁺ neuronal somata.** The manual annotators were instructed to identify every CB⁺ cell body rather than to differentiate between putative subtypes (Fig 1A). The annotation was performed using an in-house web-browser-based interface which enabled an annotator to flexibly browse the microscopic resolution images, place counting boxes, and identify individual neuronal somata. Neuroanatomists could then manually annotate the location of every neuron within a counting box by placing a circular marker centered on the soma, and then adjusting the marker radius to reflect the size of the neuronal body (Fig 2A).

In case CJ1741, following reconstruction and registration (see *Three-dimensional reconstruction*, below), columns of counting boxes, each 150 μm × 150 μm, were defined for each of the 116 cortical areas [44] to encompass all cortical layers and a fragment of underlying white matter (Fig 1B and 1C). Further, additional boxes were placed liberally to increase the diversity of the image features (e.g. blood vessels, artifacts, see Fig 1D and 1E) to improve the training of the network. This dataset, comprising 2,220 counting boxes was split randomly into training (1,617 counting boxes) and validation (603 counting boxes) batches.

The training dataset included boxes from at least one strip for each of the 116 areas from case CJ1741 (Fig 1B). These were further complemented with an auxiliary dataset comprising boxes covering the subcortical white matter and parts of the images that did not depict brain tissue (1,812 boxes in total). These boxes were inspected visually to ensure they contained no

CB$^+$ neurons (Fig 2A, bottom middle and bottom right). Subsequently, they were included in training dataset to suppress false positive rate by presenting the neural network examples of cell-alike objects such as dust and spots, smudges, etc.

In addition, in all three examined hemispheres, in nine areas selected to represent various types of cytoarchitecture (V1, V2, MT, AuA1, A3b, A3a, A4ab, A8C, and A13M; Fig 2C), additional strips were defined and annotated independently by three experts. This dataset was used to assess the performance of the automated method against multiple human raters (see *Comparison against multiple human raters*, below).

**Automated detection of CB$^+$ neurons.** The automated estimation of densities of CB$^+$ neurons relied on the *regression* or *density counting* approach [40,84,85]. This paradigm allows for estimating the *total number* of labeled cells within a defined region (e.g. counting box) of the microscopic images, in contrast to identifying individual instances of cells. Since the *regression counting* has a statistical nature, the number of detected cells might be expressed as a non-integer value. Our solution uses the U-Net [39] convolutional neural network (U-Net CNN), derived from [40] implementation, to map an input microscopic, color (24 bits per pixel, RGB) image of an immunohistochemically stained tissue into a density map of CB$^+$ neurons (Fig 2B).

For each 150 μm × 150 μm (302 px × 302 px) counting box, a corresponding ground truth density map image was generated based on markers placed by the human annotators (Fig 2B). Each marker was converted into a 2D Gaussian blob of a size proportional to the radius of the marker. This way, the entire blob sums up to 1.0 (a single neuron); hence the sum of all blobs in a counting box corresponds to the total number of neurons identified within this box. To avoid edge effects and to increase the training performance, the counting box image and the density map were cropped to 256 px × 256 px (127 μm × 127 μm), hereafter referred to as *image patches* (Fig 2A and 2B).

In summary, the training dataset comprised 3,429 pairs of image patches and corresponding density maps, of which 64.5% (2,216) contained no CB$^+$ cell annotation. In the remaining 1,213 boxes, the median number of neurons was 4.9, while the total was 9,219. The validation dataset consisted of 603 pairs of image patches and density maps totaling 3,072 neurons (a median of 4.78 neurons per patch), while 109 density maps contained no CB$^+$ cell annotation.

**Training of the U-Net CNN.** Each of the red, green, and blue channels of an image patch was independently normalized to zero mean and a unit standard deviation. The normalized image patches and corresponding density maps were then piped into four-level-deep U-Net CNN. On each level, the convolutional layers were set to 64 filters, 3 px kernel size, and ReLu activation, while the final convolutional layer was set to linear activation. The model was trained for 160 epochs with varying learning rates: $10^{-2}$ (epochs 1–40), $10^{-3}$ (epochs 41–80), $10^{-4}$ (epochs 81–120), and $10^{-5}$ (epochs 121–160). Image augmentation included intensity and contrast variations, 0–45˚ rotation, and random horizontal and vertical flips. As the purpose of the network is to map (regress) an image into a density map, the mean square error (MSE) was used as the loss function, and the mean absolute error (MAE) was additionally calculated for monitoring.

The U-Net CNN returns a total number of CB$^+$ neurons within a given image patch (Fig 2B). To calculate the density (i.e. number of objects per unit of physical volume), the following conversion is applied: $d = 0.801 \cdot n \cdot s^{-2} \cdot t^{-1}$, where $d$ is the density, $n$ is the number of neurons detected within an image patch, $s^2$ is the surface of a counting box (127 μm × 127 μm), $t$ is the nominal thickness of the section (40 μm), and the factor of 0.801 is applied to correct for the shrinkage [33].

**Training results.** The network successfully learned to map the images into CB$^+$ density maps. A strong linear relation between the ground truth and U-Net CNN-based counts ($n_m = 1.021 \cdot n_{CNN} + 0.299$, $R^2 = 0.896$) can be observed (S3A Fig), and the distribution of the

residuals averages to zero. The relative difference between the automated count and the baseline decreases with the density (S3B Fig). The CNN learned well to avoid spurious objects as the median error for a validation box that is known to contain no cells is only 0.06 (i.e. given an empty image patch, the U-Net CNN estimate is going to be $<70$ cells $\cdot$ mm$^{-3}$ for the half of the patches, S3C Fig).

## Comparison against multiple human raters

Since the model was trained on samples derived from a single marmoset hemisphere (CJ1741) manually annotated by a single expert neuroanatomist, there could be a risk of overfitting the model, which could negatively affect the performance on the hemispheres that did not contribute to the training process. To assess the model's accuracy on samples from the other cases (CJ200L and CJ205) and to confront the U-Net CNN performance against multiple experts, we generated a holdout (benchmark) dataset. In all three analyzed hemispheres, nine cortical areas (V1, V2, MT, AuA1, A3b, A3a, A4ab, A8C, and A13M; Fig 2C) were selected, thus comprising a variety of cytoarchitectural types. In each area, a single strip was defined and then annotated independently by three neuroanatomists. Therefore, the benchmark dataset amounts to 313 counting boxes per expert (939 in total). The counting boxes were pre-processed and converted into benchmark density maps as described above (Fig 2B) and were processed by the U-Net CNN. Finally, the neuronal densities were calculated based on both sets of density maps computed with the U-Net CNN and those defined by the experts.

A clear linear relation between the ground truth and CNN-based counts was found ($d_M$ = 1.093$\cdot$d$_{CNN}$—0.90, R$^2$ = 0.938, S3D Fig). The U-Net CNN results seem slightly but systematically lower than the manual estimates (95% CI: [1.07–1.11]). Analysis of the discrepancy against the neuronal density (S3E Fig) reveals that the primary sources of the differences are the counting boxes with the highest densities ($>70\cdot10^3$ cells per mm$^{-3}$, see also the red rectangle in S3D Fig), found in layer 4 of areas V1 and V2. For these densities, the automated results can be noticeably lower; for the remainder of the boxes, the automated and manual results match well.

The results also show that the densities obtained by human experts carry a noticeable variability (e.g. Fig 2D). Since we consider each expert's results equally important, we averaged ($\bar{d}_M$) the densities obtained by each human annotator ($d_{M1}$ to $d_{M3}$) for each counting box, and analyzed those densities against the U-Net CNN. First, we found that the mean U-Net CNN density for each area (i.e. the average density of all counting boxes in a given cortical area) and its manual counterpart are statistically indifferent in either absolute or relative terms (S3F and S3G Fig white violin plots). This is also the case when it comes to the individual counting boxes. However, the dispersion of the differences is much more prominent (S3F and S3G Fig violin plots in gray).

In the case of per-area estimates, the distribution of differences between the manual and the U-Net CNN densities was normal in either absolute or normalized terms (Shapiro–Wilk test: p = 0.22 and p = 0.79, respectively), and the average difference was indifferent from zero (t-test p-values of 0.13, and 0.11 for absolute and normalized differences, respectively). As far as image patch (127 μm × 127 μm) densities were concerned, the distribution of both the absolute and normalized differences was not normal (Shapiro–Wilk test: p$<10^{-5}$ in both cases), yet the differences were statistically indifferent from zero (Wilcoxon signed-rank test p = 0.12 and p = 0.07, respectively). In other words, the variance of the differences was noticeable, yet, on average, the U-Net CNN and manual results were comparable. When several counting boxes were considered (e.g. a set of boxes that belong to a single cortical area), the variance of the differences was noticeably smaller (Levene test p-values of 0.016 and $6\cdot10^{-4}$ for differences of variances between the differences of areas and boxes, absolute and normalized, respectively), with

the differences still statistically indifferent from zero. In conclusion, the U-Net CNN estimates of $CB^+$ neurons reflected the average of human experts well, particularly when an area equivalent to several counting boxes was considered (~0.25 mm$^2$, or more).

**Three-dimensional reconstruction.** To segment the individual hemispheres into cortical areas, we performed a computational alignment and three-dimensional reconstruction following procedure detailed in Figs 4 and 5 in [86] and S5 and S6 Figs in [35]. As a reference, we used the Nencki-Monash marmoset brain template [32] (RRID:SCR_018367), which represents an average morphology of twenty young adult individuals of similar age range as the individuals used in this study. This helps, to a large degree, to mitigate the issue of interindividual variability. Since the Calbindin-stained sections were used instead of Nissl-stained ones, we introduced minor changes to the procedure. In essence, a series of images of CB-stained sections covering an entire brain hemisphere (from 146 to 156 sections, depending on the case) were downsampled to a resolution of 15 μm per pixel. Next, the parts of the image representing brain tissue of a single hemisphere were selected, while the remaining voxels (contralateral hemisphere and the cerebellum) were discarded. The masking procedure was conducted using the open-source ITK-SNAP 3.8.0 application ([87], http://itksnap.org; RRID: SCR_002010). Subsequently, the sections underwent a series of two-dimensional affine alignments to each other and to corresponding template cross-sections to obtain a rudimentary reconstruction that matches the template brain hemisphere and accounts for deviations from the exact coronal sectioning plane. The affine alignment was followed by deformable corrections to refine the reconstruction and make it more suitable for the mapping into the reference template (S5B Fig).

The registration to the Nencki-Monash template [32] was carried out by the Advanced Normalization Tools (ANTs) software suite ([88], http://stnava.github.io/ANTs/; RRID: SCR_004757) with parameters akin to those specified in [86] and [35]. The non-linear registration utilized the Symmetric Normalization algorithm (SyN) with a default gradient step of 0.25. The velocity field was regularized with a Gaussian kernel (standard deviation of one voxel), and smoothing of the displacement field was turned off. The 3D reconstructions and the atlas image were resampled to an isotropic resolution of 100 μm. The registration was simultaneously driven by three pairs of images (S5E–S5G Fig) and corresponding image similarity metrics to mitigate issues related to the cross-modal (CB to Nissl) nature of the mapping.

First, grayscale (red channel) images of stained sections were registered using the cross-correlation metric ([89], CC, S5E Fig) with a kernel size of 4 voxels and a relative weight of 0.2. The second pair of images comprised label maps of cortical areas outlined manually on Nissl sections (adjacent to corresponding CB-stained sections, S5A Fig) by an expert (M.G.P.R), based on both cyto- and myelo- architecture, and only then transferred onto the CB-stained sections. An analogous set of label maps was defined in the template (S5B Fig). Subsequently, corresponding labels were forced to match during registration using the Point-Set Expectation image similarity metric ([89], PSE, S5F Fig) set to an exhaustive (100%) sampling of the labeled voxels and a relative weight of 1.0. The set of label maps delineated in each case differed slightly; see S3 Table for a detailed list. Finally, the third pair of images were maps of normalized cortical thickness computed using a 2-surface Laplacian-based approach [32,90] to constrain the displacement in the direction perpendicular to cortical lamination. The maps (S5G Fig) contained values between zero (pial surface) and one (border between gray and white matter) and were registered using the mean squared difference similarity metric (relative weight: 1.0). The results were then investigated visually and no gross misregistration or unrealistically strong displacement was identified. Finally, we assessed the post-registration overlap between the experimental case and the template for each defined label map (S5H Fig and S3 Table).

**Segmentation into cortical areas and layers.** The 3D reconstruction procedure resulted in a spatial mapping between the experimental dataset's coordinate system and the reference template's stereotaxic coordinates. We then used these transformations to map the segmentation into cortical areas from the template onto the experimental cases (S5D Fig, left).

Segmentation of layer 4 on CB-stained sections was carried out manually on every other section in each hemisphere using adjacent Nissl-stained sections as a reference (S5D Fig, middle). The border between layers 1 and 2 was defined computationally as the steepest increase of the $CB^+$ neurons density starting from the pial surface. This resulted in the segmentation of the cortex into supragranular layers (layers 2 and 3), granular cell layer (layer 4), and infragranular layers (layers 5 and 6).

In addition, to account for occasional tissue distortions (tears, folds, ruptures, etc.) and local staining artifacts, a separate mask was introduced to exclude these regions from any quantitative analyses. The mask was defined manually by closely inspecting individual sections for any of the mentioned defects.

**Whole-brain $CB^+$ density maps.** The $CB^+$ neurons density maps were generated for all sections in all cases. They were then downsampled from the native resolution (0.4974 μm per pixel, ×20 magnification) to a mesoscale resolution of 40 μm per pixel. Subsequently, they underwent spatial transformations computed in previous steps. During this procedure, the density maps were corrected by the Jacobian determinant of affine and deformable transformations to maintain accurate values (i.e. the per-section sum of all $CB^+$ neurons on the microscopic resolution maps and the spatially transformed maps are preserved). With the 1) density maps of all $CB^+$ neurons, 2) segmentation into cortical areas, and 3) segmentation into layers, it was now possible to calculate the average densities for each area and its laminar divisions (S5D Fig, left) by computing a voxel-wise average density within a mask created by intersecting relevant areal and laminar segmentations.

We excluded the amygdalopiriform transition area (APir) from the analyses due to its small size and difficulties in identifying the precise boundaries of this area, either manually or by the registration algorithm. In addition, we decided to exclude the entorhinal cortex (Ent), piriform cortex (Pir), area 24a, (A24a), area 29a-c (A29a-c), and area 35 (A35) from any analyses involving laminar divisions due to the lack of a clearly defined layer 4 homolog.

## Quantification and statistical analysis

ANOVA analyses presented in Figs 3 and 7 were carried out using the scikit-posthocs Python package (v. 0.7.0) using the Kruskal–Wallis H test. Statistical significance was assessed according to post-hoc Dunn's test with Holm–Bonferroni correction for multiple comparisons.

## Supporting information

**S1 Fig. Co-localization of $CB^+$ staining with NeuN and GABA.** Confocal images for colocalization of calbindin-positive ($CB^+$) neurons with neuronal marker (NeuN) and marker for inhibitory neurons (GABA) taken from primary motor cortex (A4ab). Color-coded arrows point to several $CB^+$ neurons, including an example of a neurogliaform neuron (yellow arrow). Scale bar: 50 μm. Primary antibodies used to create the images are: Calbindin D28K, (CB, 1:100 from Thermo Fisher, Cat# PA1-931, RRID: AB_2068509), Neuronal marker (NeuN, 1:700 from Millipore, Cat# MAB377, RRID: AB_2298772), gamma aminobutyric acid (GABA, 1:500 from Sigma-Aldrich, Cat# A2052 RRID: AB_477652). For fluorescence staining, sections were incubated in blocking solution (0.3% Triton-X100 and 10% horse serum in 0.1 phosphate buffer solution) for 1 h at room temperature, followed by 46–48 h incubation in primary antibodies. The secondary antibodies [1:600, Alexa Fluor 488 (ab150109), Alexa Fluor

594 (ab150064) and Alexa Fluor 647 (ab150111)] were applied for 60 min at room temperature.
(TIF)

**S2 Fig. Laminar density of CB⁺ neurons.** (A) Supragranular layers 2 and 3. (B) Layer 4. (C) Infragranular layers 5 and 6. Each column shows maps for the average of the three cases (top row) followed by the three individual marmosets. For each map, the scale (bottom right) indicates densities in neurons·mm⁻³.
(TIF)

**S3 Fig. Detailed evaluation of the automated cell counting method.** (A) The evaluation of the automatic cell counting performance using the validation dataset (603 counting boxes). The relation between the automated count ($n_{CNN}$, abscissas) and the ground truth manual count ($n_M$, ordinates) shows a linear relation between the two quantities. The residuals average to zero (two-sided t-test, $t < 10^{-4}$, p = 1, $\mu = 0$). (B) Relative error of the U-Net CNN ($d_{CNN}$) neuronal densities against those established by manual counting ($d_M$). Black points represent results for the individual counting boxes. Order statistics (median: thick black line, the light gray bands: 5th and 95th centiles, dark gray: lower and upper quartiles) calculated locally within a $5 \cdot 10^3$mm⁻³ wide moving window. The red dashed line represents the agreement between the U-Net CNN and the manual results. The relative error decreases with increasing CB⁺ densities (the bands are wider for lower densities and narrower for higher ones). (C) Histogram of the residual densities (i.e. the density estimated by the U-Net CNN within image patches known to contain no neurons). Among the 2,216 empty image patches, 50% had a residual density less or equal to 70 mm⁻³, and the mean residual density amounted to 170 mm⁻³, which shows that the impact of the background on the U-Net CNN results is negligible. (D) Comparison of the CB⁺ densities estimated by the U-Net CNN against three expert neuroanatomists. Densities established by the U-Net CNN ($d_{CNN}$, abscissas) against the count obtained manually ($d_M$, ordinates) for each image patch in these areas (939 values in total) show a linear relation between the two quantities. (E) Relative error of the U-Net CNN ($d_{CNN}$) neuronal densities against manual counting ($d_M$) for the benchmark dataset. Black points represent results for the individual image patches. Order statistics (median: thick black line, light gray bands: 5th and 95th centiles, dark gray: lower and upper quartiles) calculated locally within a $5 \cdot 10^3$ mm⁻³ wide moving window. The dashed red line represents the agreement between the U-Net CNN and the manual results. The densities based on U-Net CNN match those established by manual plotting for all counting boxes except those with the highest densities ($>70 \cdot 10^3$ mm⁻³), such as those obtained in the granular layer of V1 and V2. For these, the U-Net CNN densities tend to be lower than those established manually, likely reflecting occurrences of multiple, superimposed small cells. (F, G) Analyses of the discrepancies between the $d_{CNN}$ and the $\bar{d}_M$ densities, either for absolute (F) or normalized (i.e. divided by the average of the three expert observers, $\bar{d}_M$) (G) density values. In either approach, the mean difference is statistically indifferent from zero. However, per-box comparisons (gray violin plots) exhibit higher variance than the per-area comparisons (white violin plots).
(TIF)

**S4 Fig. Evaluation of the automated cell counting method against staining intensity.** Relation between the mean tile intensity of the validation (panel A) or the benchmark dataset (panel B) and the relative error of the U-Net CNN ($d_{CNN}$) neuronal densities against those established by manual counting ($d_M$) presented as relative difference: $(d_{CNN} - d_M) / (d_M + d_{CNN})$. The mean tile intensity is based on all three color channels and then normalized to the 0–1 range. Black points represent results for the individual counting boxes. Order statistics

(median: thick black line, the light gray bands: $5^{th}$ and $95^{th}$ centiles, dark gray: lower and upper quartiles) calculated locally within a 0.05 wide moving window. The red dashed line represents the agreement between the U-Net CNN and the manual results. The thick blue curve represents the fraction of tiles of equal or lower mean intensity (cumulative distribution). The up (↑) or down (↓) arrows on the abscissa axis represent the mean tile intensity values for which examples of image tiles are provided in panels C (for the validation dataset) and D (for the benchmark dataset). The relative error increases with mean tile intensity (the bands are wider for higher mean intensity and narrower for lower), however, it remains relatively stable up to the average image intensity of approximately 0.75, which corresponds to approximately 80% of image tiles for which the relative error could be described as independent from the mean image intensity. It has to be noted that the mean tile intensity of 0.75 and higher (gray level of 192 in 8-bit images) corresponds mainly to bright image tiles (see the bottom center and the bottom right examples in panels C and D) mainly from the bottom of the strips (e.g., deep layer VI or white matter). Such tiles typically have only a few cells which exaggerates the relative error due to the division by small numbers. Overall, these findings suggest that the U-Net model is similarly affected by staining variation as the manual plotting.
(TIF)

**S5 Fig. Cytoarchitecture-aware registration to Nencki-Monash reference template [32].**
(A) Label maps outlined manually on Nissl sections (*left*) are transferred onto the adjacent CB-stained sections (*right*). (B) Registration of the experimental case to the reference template. A combined view of the CB-stained sections and the outlined label maps (*left*) in the experimental case, the Nissl-stained section, and the label maps in the reference template (*right*). (C) The computed spatial transformations are used to map the segmentation from the reference template (*right*) onto the experimental case (*left*). The black line indicates the coronal location of the section shown in panels A and D–G. (D) Example CB-stained coronal section segmented into cortical areas (*right*), manual segmentation of the same section into supragranular, granular, and infragranular layers (red, green and blue, respectively, *middle*). The combination of the areal and laminar segmentation allows for computing densities of $CB^+$ neurons in individual areas across different layers (*left*). See S1 File for a full list of areas, color coding and abbreviations. (E–G) Pairs of images and respective metrics used to simultaneously drive the registration. (E) Cross-correlation (CC, [89]) between grayscale images of the stained sections, here shown in color for clarity. (F) Point-Set Expectation (PSE, [88]) metric forces corresponding label maps from the experimental case and the template to overlap. See S3 Table for a detailed list of label maps delineated for each case. (G) Normalized cortical thickness maps calculated using a 2-surface Laplacian-based approach [32,90]. Zero (blue color) corresponds to the pial surface, while one (red) to the border between the gray and the white matter. (H) Registration accuracy expressed with the Dice coefficient between individual label maps. The violin plots represent distributions of the overlap values between pairs of corresponding maps. The thick black lines represent the median values of the Dice coefficient for all label maps in each case. See S3 Table for registration accuracy for individual label maps.
(TIF)

**S6 Fig. Additional insights into data quality and individual variation in the density of $CB^+$ neurons.** A) Relation between the ratio of voxels annotated as artifacts against the total number of voxels per area in all three brain hemispheres. For clarity, data for all 345 areas are plotted with a thick black line. Areas with more than 5% of their volume affected by artifacts are displayed with color-coded points and annotated. Out of all areas, over 35% did not suffer from a single artifactual voxel, while only slightly over 4% of areas (14 out of 345) had more than 5% of their volume annotated as artifacts. In other words, almost 96% of areas had fewer

than 5% of voxels excluded from analysis. B) Linear relation between an area's mean CB$^+$ neuronal density and its standard deviation. The points represent the mean density per area in each case (red–CJ1741, green–CJ200L, blue–CJ205), the color lines represent the best linear fit for each case, and the dashed black lines indicate the 95% confidence interval for the fit. The results are consistent among the three cases and indicate a strong dependence of the dispersion on the mean value of a sample (i.e. heteroskedasticity). This implies caution when using the standard deviation to interpret the variance of densities. C-E) Maps of the coefficient of variation (ratio of the standard deviation to the mean, CV = $s_i$ / $\bar{d}_i$) for all areas in three animals reveal no clear patterns of the degree of variation.
(TIF)

**S1 Table. Source dataset for reproducing analyses presented in Figs 3–7.**
(XLSX)

**S2 Table. Source dataset for analyses of the variance in densities.**
(XLSX)

**S3 Table. Source dataset for valuation of the registration accuracy.**
(XLSX)

**S1 File. Reference card with the full names, abbreviations, color-codes and flatmap locations of the individual cortical areas, groups of areas, and the cytoarchitectural categories of lamination.**
(PDF)

## Acknowledgments

The authors would like to acknowledge the contributions of Cecilia Cranfield, Daria Malamanova and Melissa Chong during the data input phase (manual annotation of images). We also thank Emilia Chojak and Piotr Szulim for assistance in data quality assurance and image processing.

## Author Contributions

**Conceptualization:** Marcello G. P. Rosa, Piotr Majka.

**Data curation:** Nafiseh Atapour, Shi Bai, Piotr Majka.

**Formal analysis:** Nafiseh Atapour, Marcello G. P. Rosa, Sylwia Bednarek, Agata Kulesza, Gabriela Saworska, Sadaf Teymornejad, Piotr Majka.

**Funding acquisition:** Nafiseh Atapour, Marcello G. P. Rosa, Piotr Majka.

**Methodology:** Nafiseh Atapour, Marcello G. P. Rosa, Katrina H. Worthy, Piotr Majka.

**Project administration:** Marcello G. P. Rosa, Katrina H. Worthy, Piotr Majka.

**Software:** Shi Bai, Sylwia Bednarek, Piotr Majka.

**Supervision:** Marcello G. P. Rosa, Piotr Majka.

**Validation:** Nafiseh Atapour, Marcello G. P. Rosa, Piotr Majka.

**Visualization:** Agata Kulesza, Gabriela Saworska, Piotr Majka.

**Writing – original draft:** Nafiseh Atapour, Marcello G. P. Rosa, Piotr Majka.

**Writing – review & editing:** Nafiseh Atapour, Marcello G. P. Rosa, Shi Bai, Sylwia Bednarek, Piotr Majka.

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
