## [Decision Letter · Decision Letter 0]

22 May 2024

Dear Mr Majka,

Thank you very much for submitting your manuscript "Distribution of calbindin-positive neurons across areas and layers of the marmoset cerebral cortex" for consideration at PLOS Computational Biology.

As with all papers reviewed by the journal, your manuscript was reviewed by members of the editorial board and by several independent reviewers. In light of the reviews (below this email), we would like to invite the resubmission of a significantly-revised version that takes into account the reviewers' comments.

The reviewers were very enthusiastic about your work, congratulations. Indeed, their recommendations were labeled as "minor", and that seems to be the case one by one. Nevertheless I'm putting "major revision" because the list was somewhat long.

We cannot make any decision about publication until we have seen the revised manuscript and your response to the reviewers' comments. Your revised manuscript is also likely to be sent to reviewers for further evaluation.

Sincerely,

Lyle Graham

Section Editor

PLOS Computational Biology

Reviewer's Responses to Questions

**Comments to the Authors:**

Reviewer #1: The present study examines the 3D distribution of calbindin-positive (CB+) neurons across the entire marmoset cortex.

It uses immunohistochemistry for imaging the CB+ neurons, and proposes a workflow consisting of automatic cell density estimation and 3D reconstruction of histological sections to extract the density distribution and compute quantitative maps organized by cytoarchitectonic structures.

The approach offers valuable novel insights in the cytoarchitecture of the primate brain, revealing distinct patterns of CB+ neuron distribution across various cortical areas and layers, as well as the relationship to general neuronal densities.

The reported results should be of significant interest to the community.

The experiments are based on serial sections of one hemisphere of each of three young adult marmoset brains, where immunohistochemistry stains are applied to each 5th section. Of the remaining sections, adjacent ones were stained for NeuN, myleon and cytochrom oxidase to identify areas and layers.

Quantification of cell densities is based on a conventional U-Net model which is trained to predict cell numbers of image patches in the form of sparse Gaussian kernel density maps. The network is trained from expert annotations in sample strips for each studied brain regions extracted from one of the cases. Validation of the trained model is done against expert annotations in a selection of brain areas in all three brains.

The paper is well written and organized. The experimental results are comprehensive, convincing, and well presented. The data and machine learning model are provided alongside the paper, including the labelled training dataset, which is much appreciated to reproduce the results.

However, I identified a couple of minor issues that should be taken into account to improve the manuscript:

- While the paper claims at several places (e.g. introduction, beginning of results section, top of page 15) to study all lamina of the cortex, the actually reported results refer to groups of lamina (2/3, 4, 5/6). This should be properly reflected everywhere in the text.

- I was wondering why there are no standard deviations displayed for density estimates, such as in Fig. 4 and Fig. 6. Isn't it so that densities are predicted for sets of small patches and averaged per region/region group? If yes, I suggest to display them in these figures. This would also help to interpret the variance of densities across the three cases better, which is quite significant in some cases (e.g. OFC).

- Somewhat related, I found it rather hard to compare the density values in Fig 4 between the three cases, due to the way the barplots are organized into split columns. It seems to me that if the bar plots were transposed, one could have one row per case with corresponding areas vertically aligned, then visually much closer to each other and better to read. Did you try such layout?

- While the registration workflow has been published earlier by Majka et al., I find the description and evaluation of the approach a bit short in the paper. In particular, it could be mentioned between which images the correlation metric is evaluated. More importantly, although I have no reason to assume insufficient registration quality, it would be helpful to see some quantitative assessment of the accuracy. On p. 27 it is mentioned that adding the label maps to guide the registration improves the accuracy, but I did not find any explanation how this was measured and how significant the effect was.

- I did not find an explanation about which hemispheres were used in each case, and why it was decided to only analyze one hemisphere per brain. This should be briefly explained.

- It is mentioned that tissue distortions were masked and excluded from evaluations. It would be helpful to report the average amount of corrupt tissue, and especially valuable to provide these masks together with the data.

- The discussion of alternative cell detection methods on page 20 is a bit outdated and selective. I would suggest to briefly mention the current state of the art represented by the recent NeuRIPs challenge (Ma et al., Nat Meth 2024, https://doi.org/10.1038/s41592-024-02233-6). Of course I do not expect to repeat any experiments with other instance segmentation models, since the authors do not claim to use the best available method and explicitly encourage to try other models using the published training data.

- The exluded brain areas discussed on p. 27 (bottom) should be reported with the limitations of the approach

- The introduction motivates that there are comprehensive existing resources on distributions of cortical neurons in the marmoset brain. I would have expected that at least some of these published measures are mentioned in the discussion later on.

Some additional things I came across:

- In the caption of Fig. 2, the sentence starting with "Upon training..." seems to have some misplaced commas

- I was wondering if the coloring of the brain area groupings (e.g. Fig. 3A) coul be used in the coloringn of the flat maps (e.g. Fig3D) as well. This would make it easier to interpret the figure, especially since the group names or not shown in the flat map either.

- I suggest to annotate figures consistently with the case IDs where applicable to improve the connection between them (e.g. in Figure 4 they are not given)

Reviewer #2: majka review

The manuscript of Majka et al. describes their work mapping the distribution of calbindin-positive neurones in the marmoset cerebral cortex. The result is very interesting, as it provides the first extensive map of a neuronal type in a widely used non-human primate brain. Even more interesting, their rigorous and well thought methodology should provide a template for further studies of other markers and other species. This methodology succeeds in combining input from experts (manual segmentation and annotation) with sophisticated computational neuroanatomy analyses. It is this combination which enables them to scale the focused work of experts to the whole cerebral cortex, providing analyses at the level of cytoarchitectonic areas as well as cortical layers. It is highly commendable that their data (including the annotations used for training) and their code are made openly accessible and open source.

The manuscript is well written, very clear; the illustrations are also clear and informative; and information about statistical analyses seems adequate to assess the validity of their results (interested readers will also be able to download data and code if they want to perform additional validation).

I have only one, minor, question: despite the careful approach to 3D reconstruction and estimation of neuronal densities, the map in figure 2G shows vertical coronal bands which suggest an artefact in density estimation due to the slicing plane. Could the authors discuss a bit the origin of this artefact? Would it be possible to quantify it or compensate it? Low frequency filtering could be a possibility, however, it may blur important information especially when rapid changes in density occur. Would it be imaginable to combine maps estimated from coronal sections with axial or sagittal maps?

R. Toro

**Have the authors made all data and (if applicable) computational code underlying the findings in their manuscript fully available?**

Reviewer #1: Yes

Reviewer #2: Yes

PLOS authors have the option to publish the peer review history of their article (what does this mean?). If published, this will include your full peer review and any attached files.

Reviewer #1: No

Reviewer #2: **Yes: **Roberto Toro
---

## [Decision Letter · Decision Letter 1]

16 Aug 2024

Dear Mr Majka,

We are pleased to inform you that your manuscript 'Distribution of calbindin-positive neurons across areas and layers of the marmoset cerebral cortex' has been provisionally accepted for publication in PLOS Computational Biology.

Best regards,

Lyle Graham

Section Editor

PLOS Computational Biology

Reviewer's Responses to Questions

**Comments to the Authors:**

Reviewer #1: The revision addresses all my previous remarks. I think that the article can be published in the revised form. I thank and congratulate the authors, this is a very nice paper!

**Have the authors made all data and (if applicable) computational code underlying the findings in their manuscript fully available?**

Reviewer #1: Yes

PLOS authors have the option to publish the peer review history of their article (what does this mean?). If published, this will include your full peer review and any attached files.

Reviewer #1: **Yes: **Timo Dickscheid

---

## [Editor Report · Acceptance letter]

14 Sep 2024

PCOMPBIOL-D-24-00511R1 

Distribution of calbindin-positive neurons across areas and layers of the marmoset cerebral cortex

Dear Dr Majka,

I am pleased to inform you that your manuscript has been formally accepted for publication in PLOS Computational Biology. Your manuscript is now with our production department and you will be notified of the publication date in due course.

With kind regards,

Jazmin Toth
